bioinformatics/image processing/materials science

connected components labelling, sequential region labelling, parallel processing

**Author for correspondence:**
Michael Doube
e-mail: mdoube@cityu.edu.hk

# Multithreaded two-pass connected components labelling and particle analysis in ImageJ

## Michael Doube

Department of Infectious Diseases and Public Health, City University of Hong Kong, Tat Chee Avenue, Kowloon, Hong Kong SAR

(iD) MD, 0000-0002-8021-8127

Sequential region labelling, also known as connected components labelling, is a standard image segmentation problem that joins contiguous foreground pixels into blobs. Despite its long development history and widespread use across diverse domains such as bone biology, materials science and geology, connected components labelling can still form a bottleneck in image processing pipelines. Here, I describe a multithreaded implementation of classical two-pass sequential region labelling and introduce an efficient collision resolution step, 'bucket fountain'. Code was validated on test images and against commercial software (Avizo). It was performance tested on images from 2 MB (161 particles) to 6.5 GB (437 508 particles) to determine whether theoretical linear scaling ($O(n)$) had been achieved, and on 1–40 CPU threads to measure speed improvements due to multithreading. The new implementation achieves linear scaling ($b = 0.905$–$1.052$, $time \propto pixels^b$; $R^2 = 0.985$–$0.996$), which improves with increasing thread number up to 8–16 threads, suggesting that it is memory bandwidth limited. This new implementation of sequential region labelling reduces the time required from hours to a few tens of seconds for images of several GB, and is limited only by hardware scale. It is available open source and free of charge in BoneJ.

## 1. Introduction

Region labelling is a standard segmentation problem in two-dimensional (2D) and three-dimensional (3D) image processing, in which groups of pixels that form contiguous domains are identified and each discrete domain is given a unique label [1]. Region labelling is needed across disciplines, for example, to identify pores in sintered polymers [2], soil [3], rock [4,5], cement [6], noodle dough [7], teeth [8] or bone [9], from images

obtained by modalities such as X-ray microtomography (XMT), confocal microscopy or serial sectioning microscopy. It is also useful to perform region labelling to remove small particles from images prior to calculating the Euler characteristic of a connected structure, for example, to calculate trabecular bone's connectivity density (Conn.D), a measure of the number of trabecular struts [10]. XMT images on contemporary instruments are often $ca$ $2048^3$ pixels, or larger after stitching of projections or image stacks. Parallelized connected components labelling algorithms have existed for at least the last 30 years [11,12], yet slow region labelling can still be an obstruction to measuring features of interest in a timely manner, especially on large 3D image data.

Various multithreaded, chunked, single-threaded and mapped region labelling algorithms were included in BoneJ [13] from its earliest releases and improvements made in subsequent years. The starting point for these algorithms was the 3D Object Counter ImageJ plugin [14] published open source by Cordelières and Jackson (reported in [15]). The 3D Object Counter was a naive and recursive implementation that scaled as approximately $O(n^2)$ because every time a label change occurred, all the image pixels were iterated to update matching labels. For small images on the order of a few tens of MB, processing times were tolerable, but for larger images of hundreds of MB and above, processing times became prohibitive.

Alternative approaches that avoid recursion such as *breadth-first* and *depth-first* flood fill algorithms [1] record the coordinates of foreground pixels and may function adequately for small images, but used on large images can consume an excessive amount of memory. In these algorithms, every 1-byte foreground pixel is represented by 12 bytes of coordinate data (3 dimensions × 4-byte integers), which can easily exceed available system memory for larger images. More complex two-pass approaches record collisions between neighbouring pixel subregions during the first pass, resolve the collisions in an intermediate step and merge connected subregions in the second pass. The network of collisions may be visualized as a graph, with subregion labels as nodes and the collision as an edge connecting two adjacent subregion labels [1]. Because each pixel is accessed a fixed number of times, two-pass approaches have the potential to scale linearly (as $O(n)$) and thus perform well on large data, provided that label collision resolution is efficient. Improvements to linear speed should also be possible by multithreading the two-pass algorithm; however, collisions between threads during label collision recording and resolution require either synchronization of data access or careful design to ensure threads never read and write the same data at the same time.

This report proposes a multithreaded implementation of two-pass sequential region labelling with efficient collision resolution and tests the hypotheses (i) that linear scaling ($O(n)$) is achieved and (ii) that further linear speed improvements may be achieved as the number of processor threads increases.

## 2. Installation and code availability

The current implementation has been developed in the Java programming language for the popular scientific image processing platform ImageJ [16]. The simplest and recommended way to install the software is by installing the Fiji Is Just ImageJ (Fiji) [17] bundle from https://fiji.sc/, and adding the BoneJ update site [18] to Fiji according to the instructions at https://imagej.github.io/BoneJ. This installs the command `Particle Analyser` in the user menus, which can be found using the keyboard shortcut [L] or using the search bar in the ImageJ GUI.

Code changes are tracked using Git [19] and published at GitHub (https://github.com/bonej-org/BoneJ2), while release version code is archived at Zenodo under a BSD 2-clause licence (see Data accessibility). In BoneJ1, all methods were contained in a single class, `ParticleCounter`. The current release achieves separation of concerns by creating three new classes: `ConnectedComponents` (labels regions); `ParticleAnalysis` (measures region features); `Particle Display` (handles graphical output); leaving `ParticleCounter` as a 'master' plugin that coordinates user input and output with ImageJ's API and the three new classes. `ConnectedComponents` is intended to be reused by other ImageJ plugins via its public `run()` method.

## 3. Description of the algorithm

### 3.1. Input data

The algorithm expects a 2D or 3D image that has been segmented into foreground and background. In ImageJ this is an 8-bit binary image that contains only 0 (background) and 255 (foreground). Other

numbers of dimensions could conceivably work with the current approach, provided that suitable neighbourhoods and multithreading chunking strategies are devised.

## 3.2. First pass

The first pass proceeds as described in the classical algorithm [1] with some important variations. An ID value is initialized with a value of 1 to increment region labels. Label 0 is reserved for background. A neighbourhood that reads only the previously visited pixels iterates through all the pixels in a raster pattern, labelling the current foreground pixel with the smallest label value found among the neighbouring pixels. Only the 4 pixels new to the 13-neighbourhood are read from the label image and the other 9 pixels are reused by shifting them one to the left in the neighbourhood array. If no smaller label than ID is found, the current pixel is set to ID and ID incremented by 1. Label pixels are stored in a primitive integer array that has the same dimensions as the input image (figure 1*a*).

In preparation for later collision resolution, the neighbourhood pixel values are added to an `ArrayList` of `HashSets`, which can be visualized as a column of numbered buckets (figure 1*b*). Java's standard `ArrayList` and `HashSet` were replaced by the more efficient `MutableList` and `IntHashSet` implementations from Eclipse Collections [20], and are referred to for the remainder of this document as `ArrayList` and `HashSet` because the functionality is the same as Java's. The position (index) of a bucket (`HashSet`) in the column (`ArrayList`) relates to a region label. The contents of each bucket are neighbouring labels. Each bucket is initialized containing one label matching the bucket's position in the column. During the first pass, the bucket with its position matching each neighbour pixel is looked up and the current pixel's label added to it. Efficiency is gained by skipping background neighbours, neighbours matching the current pixel and neighbours that are the same as the last neighbour to be added. Because the current pixel's label is always less than or equal to the neighbours' labels, buckets receive only labels that have lower values than the buckets' positions. Use of `HashSets` prevents the same label from being added to a bucket more than once. It is not necessary to store pairs of labels as nodes and edges, as described in the classical algorithm.

## 3.3. Collision resolution

The presence of labels within a bucket means that all the labels belong to the same region. Although it helps to consider the underlying graph structure of the labels in a region, there is no need to reconstruct the specific connections among labels and these individual relations are ignored. If a label is found in more than one bucket, then the contents of all buckets containing that label can be aggregated into a single bucket. The first step in this algorithm's collision resolution allows labels to flow downwards into lower buckets, merging buckets' contents on the way, which is somewhat like the flow of water in the kinetic sculpture *Bucket Fountain* by Burren & Keen [21] installed in Cuba Mall, Wellington, New Zealand. Starting from the highest bucket and proceeding to the lowest bucket, all the labels in each bucket are checked (figure 1*c–f*). If labels are found that are lower than the bucket's position, the bucket is emptied into the lower bucket whose position equals the lowest label in the upper bucket. Later in the iteration, the lower bucket will have all its labels checked, and all of the labels that have accumulated in it from higher buckets may be emptied into another, lower, bucket. Implementing buckets as `HashSets` means that each time a bucket pours its labels into another bucket by the `HashSet.addAll(HashSet)` method, redundant labels are eliminated. In the final state, all the buckets have been checked and the labels in a bucket are all greater than or equal to the bucket position, which is the opposite of the starting state.

The *bucket fountain* does not always merge labels that are transitively connected (figure 1*f*). A second step checks consistency by iterating back up through the buckets ensuring that labels are always bigger than the bucket number, and that each label appears exactly once within the whole set of buckets. If either condition is not satisfied, the offending bucket is emptied as before, into the bucket matching the lowest label. Consistency is checked in this manner in a `while()` loop, until no inconsistencies are found, typically with two and sometimes three iterations. During this process, labels are paired with the lowest label in their bucket, and a look-up table (LUT) is constructed that translates the first pass label to a final label. Each bucket (`HashSet`) is paired with its minimum label in a `HashMap`. A second `HashMap` associates first pass labels and the minimum replacement label. The `hashMap`'s set of first pass labels is checked against the replacement label stored in the `lutMap` to ensure that the first pass label is being minimized. Once the labels are minimized within buckets, the content of each bucket represents the complete set of first pass labels that represent each region. Gaps between buckets are

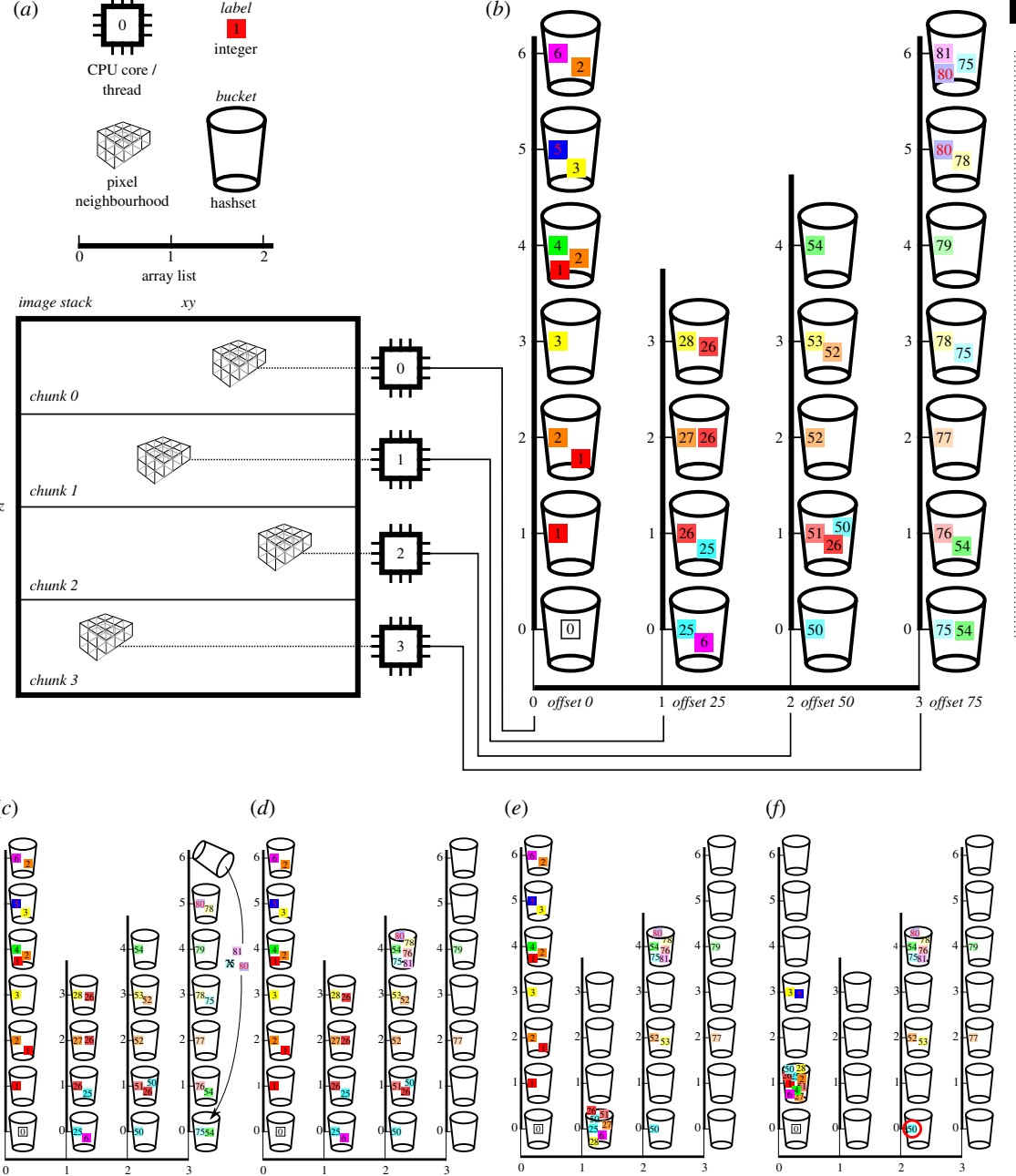

**Figure 1.** Visual summary of the first pass and of bucket fountain collision resolution. (*a*) Half-neighbourhoods, one per thread and one thread per image chunk, raster over the image pixels. (*b*) Each thread addresses its own column of buckets, adding to the bucket that matches the neighbourhood centre pixel the labels of the other neighbours, which always have lower values than the centre pixel. Each bucket always contains its own label, and is created only once needed. In this example, a total label range of 100 is split over four chunks giving each chunk a label range of 25, and offsets of 0, 25, 50 and 75, which keep each chunk's label range separate from the others' ranges. (*c*) The bucket fountain starts on the highest valued bucket and descends towards 0, pouring labels into the bucket matching the lowest label contained in the bucket. Redundant labels (here, 75) are eliminated. Intermediate steps (*d*,*e*) demonstrate the state after emptying all buckets from the 3rd column and down to the first bucket of the first column. (*f*) The state after completion of bucket fountain. Note the orphaned label 50 with red outline in column 2, which is picked up by the subsequent consistency and look-up table (LUT)-creation step. In this toy example, bucket fountain results in simplification of 40 labels spread over 19 buckets into 23 labels in seven buckets, with one orphan.

removed by assigning consecutive labels to a new `HashMap lutLut`, which is then copied to a final LUT that is implemented as a primitive integer array for efficient storage and addressing. Making a LUT from the collision resolution diverges from the Burger & Burge pseudocode [1], which looks up replacement labels from the `HashSet` during pixel iteration.

## 3.4. Second pass

In the second pass, each first pass label is read from the label array and replaced by the value held at the label's position in the LUT.

## 3.5. Multithreading

Because this algorithm contains minimal recursion through pixels there is little need for synchronization, leading to the ability to implement a multithreading strategy exploiting the representation of $xy$ image slices as independent arrays within the 2D image stack array (figure 1a,b). In the first pass, the label image is considered as a series of chunks, each chunk relating to a discrete range in $z$. Each chunk or $z$-range is processed in a separate thread. To ensure that the label space of each chunk does not interfere with the label space of the other chunks, each chunk is assigned a range of labels, calculated as the maximum label divided by the number of chunks. A special neighbourhood that does not check the previous chunk's last slice is used on the zeroth slice of each chunk, then the usual half-neighbourhood is used for the rest of the chunk. After the whole chunk is processed, the zeroth slice is processed again using the half-neighbourhood, which detects collisions between particle labels in the last slice of the previous chunk and particle labels in the zeroth slice of the current chunk.

Each chunk's collisions are stored in an independent list (`ArrayList<ArrayList<HashSet< Integer>>>`) to avoid synchronization overhead and concurrent modification of elements (figure 1b). A label offset equal to the minimum of the chunk's label space is used to relate bucket positions, which have zero-based numbering, to labels, which are label offset indexed. This allows the full range of labels to be used without creating unneeded `HashSets`. During collision resolution, labels belonging to the prior chunk are readily recognized because they are less than the chunk's label offset. Collision resolution is performed in a single thread, and the second pass is performed in multiple threads using the same chunked $z$-range strategy as in the first pass. The final LUT is a 2D `int` array, with each chunk having its own 1D LUT, and the index for the look-up value being calculated by subtracting the chunk label offset from the label being replaced.

## 3.6. Foreground versus background

This algorithm can work with any neighbourhood configuration. The current implementation assumes a 26-connected foreground. Its complementary 6-connected background is used during a particle filtering step prior to calculating connectivity, implemented as Purify in BoneJ [13,22].

## 3.7. Analysis and display

Analysis and display options are selected by the user in a single step via a GUI dialog, called on an open image by a single menu command. All user options are displayed in one set-up dialog and no further user interaction is required to generate output images and data. The intention is to minimize menu clicking and GUI interaction, which can represent a substantial time cost to users. Reproducibility and efficiency may be enhanced by recording and running a macro script, which avoids repeated GUI clicking and can be run as a batch over many input images or incorporated into a larger workflow.

Individual particles are identified by their unique label in the label array and a binary copy, limited to the particle's extent in $x$, $y$ and $z$, is used for input to BoneJ's other analysis methods. These include connectivity [22], local thickness [23], volume, surface area, moments of inertia, skeleton branch count and total length [24], and best-fit ellipsoids, as described and validated elsewhere for single structures [13]. Any 3D binary operation available in the ImageJ plugin ecosystem could readily be applied to each particle. Particles may be displayed in stacks by their label, size, thickness, or in 3D as voxel volumes or surface meshes. Analysis results, such as best-fit ellipsoids, may also be displayed as stacks or in the 3D viewer [25].

# 4. Validation and performance testing

Simple validation was performed with a test script that generated a stack containing cubes, some with voids. An image of a single spiral in $xz$ passing through multiple image chunks several times (figure 2a) was used to ensure that merging discrete particle subregions across image chunks

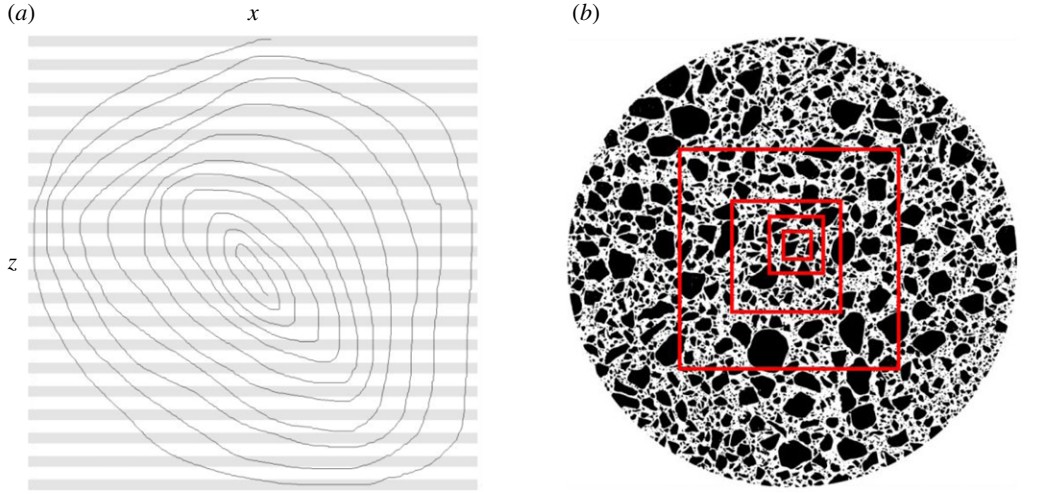

**Figure 2.** Test images. (*a*) A spiral in *xz* (image dimensions 1024 × 3 × 1024 pixels) passing through multiple chunks tests the ability of the algorithm to connect subregions that are connected through multiple transitive steps via multiple chunks. Chunks are indicated by pale grey and white horizontal stripes. (*b*) `s10up`, a binarized XMT scan of a particulate material (2103 × 2103 × 1585 pixels) that contains one large particle and 437 507 smaller particles. Smaller cubic image stacks of 1024, 512, 256 and 128 pixels wide that were used to test performance scaling behaviour are indicated on the parent volume as red squares.

completed successfully. A $7 \times 10^9$-pixel XMT image of particles (`s10up`) was obtained from the BoneJ user community and cropped into four smaller volumes, of 2, 16, 128 and 1024 MB (figure 2*b*), to cover 4–5 orders of magnitude of pixel number, with which the scaling characteristics of the algorithm were measured. All test images and scripts are available online (see Data accessibility).

The number of active CPU threads was set using the `-XX:ActiveProcessorCount=n` Java option (where $n$ = 1, 2, 4, 8, 16, 32, 40, 20, 10, 5) in combination with the ImageJ *Memory & Threads* setting, where the maximum $n$ = 40 was the number of CPU threads present in the test system (Dell T7910, 282 GB DDR4 2400 MHz RAM in 16 DIMMs, dual Intel Xeon Silver 4114 CPUs at 2.20 GHz × 20 cores/40 threads, Ubuntu 16.04, OpenJDK 1.8.0_265; Dell Hong Kong). For all tests, two to three warm-up runs were performed to allow the just-in-time (JIT) compiler to optimize code and the mean of five subsequent runs was recorded. Time $t$ to complete particle labelling for each number of active CPU cores was plotted against image size in pixels $p$, power curves ($t \propto p^b$) fitted and the coefficient of determination ($R^2$) estimated (LibreOffice Calc v. 5.1.6.2). For comparison, times were recorded for the `s10up` image series with no hardware throttling using BoneJ's previous release (v. 6.1.1) and for the current code (commit 718c7e) on a laptop (Latitude 7370, Dell UK, Ubuntu 20.04, OpenJDK 1.8.0_265).

Running code was sampled with Java VisualVM (Oracle; v. 1.8.0_221) to determine which methods were most expensive in terms of CPU time.

Each image from the *s10up* image series was opened in Avizo (v. 2020.2, ThermoFisher), and a Connected Components analysis module attached to the opened data (image.am > Image Segmentation > Connected Components) with settings as close as possible to the new code's conditions (grey image; intensity: 20–255; connectivity: corner; size: 1–0; output: label image; output type: label field (32 bit)). Performance comparisons were made on the 1 and 6.5 GB test images using BoneJ in Fiji and Avizo's multithreaded connected components module, 'Labeling', with a 3D 26-connected neighbourhood. Avizo tests and performance comparisons were performed on a Dell Precision T5820 tower with a single Intel Xeon W-2123 (4 cores, 8 threads) 3.6 GHz CPU, 64 GB 2666 MHz DDR4 RAM in 4 modules, and an 8 GB NVIDIA Quadro RTX4000 GPGPU, running Java 1.8.0_172 (Oracle) on Windows 10 (Microsoft).

# 5. Performance results

Particles are all correctly identified with no gaps between label numbers. The total number of particles in the test images agrees with the number expected, with previously validated versions of the software (figure 3*a*), and with Avizo's Connected Components module. The *xz* spiral is labelled as a single particle, demonstrating correct label collision resolution among chunks. Scaling of the complete

(a)  (b)

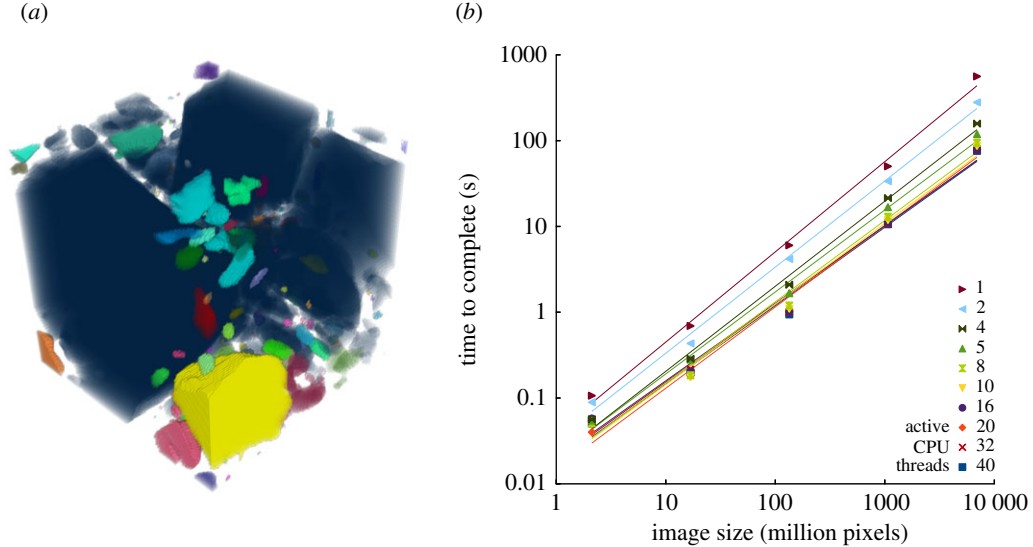

**Figure 3.** (a) Volumetric rendering of the 2 MB `s10up` test image demonstrating correct labelling of regions, including those with complex connections that span multiple chunks. (b) Performance scaling of two-pass multithreaded approach with bucket fountain collision resolution. Time to complete increases about linearly ($O(n)$) with increasing image size, and decreases with increasing number of active CPU threads. Note the slight decrease in gradient as number of CPU threads increases, indicating improving scaling to less than linear, but the overlapping of regression lines for greater numbers of CPU threads cores indicating diminishing returns above 8–10 threads.

**Table 1.** Scaling exponents ($b$), coefficients of determination ($R^2$) and completion times, calculated on the five *s10up* test images from $2 \times 10^6$ pixels ($p$) to $7 \times 10^9$ pixels, where time to complete $t = x + ap^b$. Scaling remains close to or below the theoretical optimum of $O(n)$ and in general improves as number of active threads increases. For all but the largest image, performance gains are marginal for more than 8–10 threads.

| active CPU threads | scaling $t \propto p^b$ | | time to complete (s) | | | | |
| --- | --- | --- | --- | --- | --- | --- | --- |
| | $b$ | $R^2$ | 2 MB | 16 MB | 128 MB | 1 GB | 6.5 GB |
| 1 | 1.052 | 0.996 | 0.107 | 0.699 | 6.112 | 51.12 | 574.2 |
| 2 | 1.004 | 0.996 | 0.090 | 0.437 | 4.237 | 34.27 | 285.6 |
| 4 | 0.991 | 0.995 | 0.057 | 0.289 | 2.114 | 21.81 | 161.4 |
| 5 | 0.956 | 0.995 | 0.055 | 0.283 | 1.686 | 17.05 | 121.2 |
| 8 | 0.947 | 0.988 | 0.051 | 0.185 | 1.210 | 13.13 | 95.8 |
| 10 | 0.939 | 0.987 | 0.050 | 0.176 | 1.113 | 12.08 | 88.5 |
| 16 | 0.905 | 0.985 | 0.058 | 0.206 | 1.028 | 11.38 | 79.9 |
| 20 | 0.947 | 0.992 | 0.040 | 0.183 | 1.065 | 11.32 | 80.7 |
| 32 | 0.910 | 0.987 | 0.053 | 0.217 | 1.013 | 10.94 | 80.0 |
| 40 | 0.913 | 0.986 | 0.051 | 0.201 | 0.954 | 10.78 | 77.5 |

labelling process as a function of image size is approximately linear, i.e. $O(n)$, with $b = 0.905$–$1.052$ and $R^2 = 0.985$–$0.996$ (table 1 and figure 3b). First ($b = 0.89$, $R^2 = 0.982$) and second ($b = 0.79$, $R^2 = 0.972$) passes scale as less than $O(n)$, while *bucket fountain* scales at slightly more than $O(n)$ ($b = 1.03$, $R^2 = 0.993$; figure 4). Scaling exponents less than 1 indicate that larger images may be processed at a greater rate than smaller images.

Sampling indicated that `HashSet.add()` within `addNeighboursToMap()` is the biggest single user of CPU during first labelling pass (5–20%), while 'self time' within `firstIDAttribution()` also contributes a large fraction (80–95%).

Processing speed in particles s$^{-1}$ and pixels s$^{-1}$ and overall finishing time saturates at around 8–16 CPUs on the test machine, depending on data size. The 128 MB test image was consistently fastest in

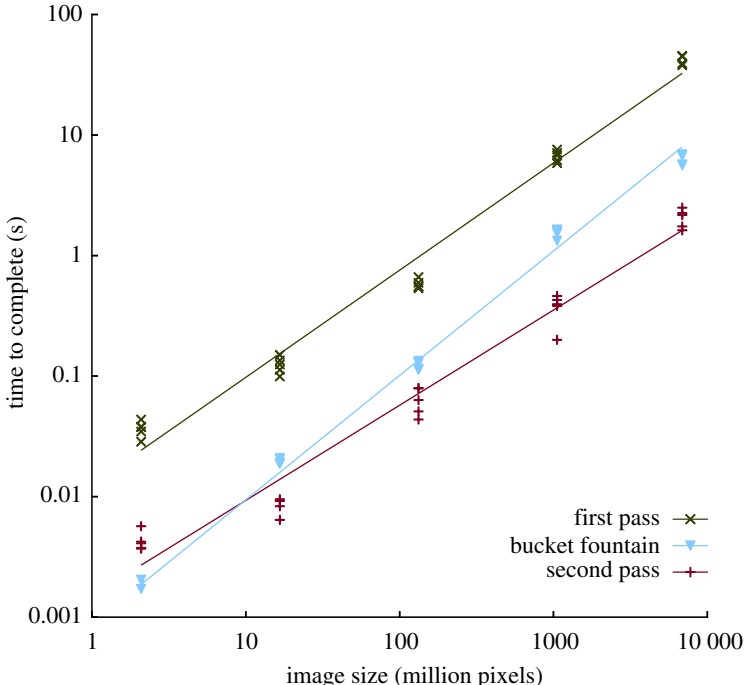

**Figure 4.** Timer code on the first pass, *bucket fountain*, and second pass sections of the implementation reveals the amount of time spent in each, tested on 40 threads. First ($b = 0.89$, $R^2 = 0.982$) and second ($b = 0.79$, $R^2 = 0.972$) passes scale as less than $O(n)$, while bucket fountain scales at slightly more than $O(n)$ ($b = 1.03$, $R^2 = 0.993$). In the second pass, a very simple pixel LUT operation is applied and may be considered a minimal time to iterate the image once. The first pass is 10–20× slower than the second pass due to multiple pixel accesses and the expensive operation of adding `HashSet` elements to the collision map during iteration through each pixel neighbourhood. Reducing the cost of building the collision map during the first pass will be the focus of further optimizations.

pixels per second, perhaps because the image array size ($512 \times 512$ pixels, 256 kB slices) neatly matches the memory configuration leading to efficient data transfer.

The new implementation is 35–60× faster than the previous release (128 MB: 57 s; 1024 MB: 422 s; 6.5 GB: 2891 s). On the laptop, completion times were tolerable (16 MB: 0.322 s; 128 MB: 2.6 s; 1024 MB: 21 s), but the 6.5 GB image caused an out-of-memory exception. Completion times were about 15× less than Avizo's multithreaded Labeling module, (1 GB: 9.5–11 s versus 165 s; 6.5 GB: 65–72 s versus 1220 s).

## 6. Discussion

This implementation of sequential region labelling achieves scaling at or better than the theoretical optimum of $O(n)$. It may be relatively more efficient on larger images than on smaller images, perhaps through an accelerated rate of complexity reduction in larger more connected particles during the first pass as it populates the `HashSets` used by *bucket fountain*, or due to Java's JIT compiler and adaptive optimization increasing data throughput of larger arrays during first and second passes. Alternatively, smaller images may incur a penalty related to constant-time features of the algorithm, such as setting up threads, which are less well amortized over their shorter sequential reads during first and second passes. *Bucket fountain* belongs to a class of algorithms known as *union-find* and is similar to the optimized array-based union-find by Wu *et al.* [26]. Here, the number of times data is retrieved from or written to RAM is minimized by accessing pixels non-recursively. Data reduction is achieved early by not storing redundant label collisions, and merging transitively connected labels efficiently with Java's `HashSet`. `HashSet.add()` is the most consuming of CPU wall-clock time, which may relate to instantiating and writing new Java `Objects`, or calculating hashes. `HashSet` operations are minimized by the *bucket fountain*, which attempts to aggregate labels with the least handling of each `HashSet`'s contents by merging to the lowest bucket possible, with the side effect that some transitively linked labels are orphaned and must be recovered by a subsequent consistency check. An alternative, more careful implementation that joins all subregions in a single pass *snowballs*

through consecutive buckets, merging each bucket's contents to the next lower (not the lowest) bucket referred to by its labels. The snowball approach makes completion times 20–100% longer than the bucket fountain in practice, presumably because each label is handled multiple times as the regions are aggregated while they pass through a larger number of buckets. Fast and sloppy with a small amount of recursive error correction notably outperforms slow and careful with no error correction in this case.

This multithreaded implementation of sequential region labelling displays memory-bound rather than CPU-bound behaviour and on the test system hits the so-called *memory wall* [27] at around 8–10 threads for smaller images of 1 GB and below. Performance would increase most readily as a function of increasing memory frequency and parallelism, which may require entirely new hardware architectures [28]. Future optimizations that reduce RAM access, for example, by reducing pixel lookups and `HashSet.add()` operations with a decision tree [26], and that improve utilization of fast CPU cache memory, may ameliorate the current bottleneck. Alternative implementations exist in programming languages that can be more efficient than Java at accessing large amounts of array data, such as C and C++ [26,29], or that store particle label collisions in primitive arrays that can be faster to create and access than the Java (Eclipse) Collections used here. This Java implementation far exceeds the performance of the commercial C++ implementation included in Avizo's Labeling module, with further optimizations still possible such as a kernel decision tree (after [26,29]). A major goal of this project was to produce an implementation that is trivial to install and run for users of the popular ImageJ platform [16]. The Fiji distribution makes it convenient for users to install and operate the plugin, and ImageJ is written in Java, hence the design decision to implement this algorithm in Java.

The redesign of the code to be more modular addresses concerns expressed in previous work, such as Mader *et al.* [30] who complained that the prior implementation in BoneJ '*lacks the flexibility to perform a number of different analyses, necessary for large-scale studies and detailed analysis of distribution, alignment, and other osteocyte lacunar measures*'. Once the label image is created, any range of analyses could be performed on the labels, by adding analytic methods to BoneJ's new `ParticleAnalysis` class, by obtaining the label image programmatically by calling the `ConnectedComponents.run()` method from client code, or by running custom analysis code such as a Python script or ImageJ plugin on the label image that is displayed in the ImageJ GUI. The label image could also be saved and opened in other software.

Contemporary computers have benefited from algorithms implemented for general-purpose computing on graphical processing units (GPGPU) via NVIDIA's CUDA library or the OpenCL library. The main advantage of GPGPU processing is the large number of parallel data streams that can be processed and the high transfer rate within video RAM (VRAM; GDDR6 at *ca* 400 GB s$^{-1}$ versus DDR4 RAM at *ca* 20 GB s$^{-1}$). It is, therefore, tempting to consider porting the current memory bandwidth-limited algorithm to exploit the GPGPU's parallel architecture. Unfortunately, at the time of writing OpenCL does not have a `HashSet` implementation, which is a core part of the label collision recording and resolution strategy. In addition, the time taken to read data from RAM into VRAM for GPGPU processing is unlikely to be substantially less than to read from RAM during CPU processing; however, this cost may be mitigated by the multiple pixel accesses during first pass labelling occurring on VRAM rather than RAM. Finally, available VRAM is typically much less than system RAM, with contemporary machines being able to hold up to 3 TB of RAM on a single motherboard (Dell 7920), but GPGPUs typically contain one or two orders of magnitude less VMRAM (NVIDIA QUADRO TX: 48 GB), limiting the maximum image size that could be handled by GPGPU compared to CPU. Given the high performance of the CPU implementation, the cost : benefit of developing a GPU implementation requires careful consideration.

Java integers are 4-byte signed values. Using 0 as the background label and 1 as the minimum label gives a range of $2^{31} - 1 = 2\,147\,483\,647$ possible labels. It is possible to double the label range if Integer.MIN_VALUE ($-2\,147\,483\,648$) was used as the background value, and labels started from Integer.MIN_VALUE + 1; however, in practice there are usually several orders of magnitude fewer particle labels required than pixels in the image: the 1 GB test image contains over $1 \times 10^9$ pixels and fewer than $1 \times 10^5$ particles (about one particle per 10 000 pixels, which is one particle per cubic box of 21.5 pixels edge length). A practical consideration is that ImageJ displays integer arrays as RGB, which is unhelpful for later analysis: to avoid this, the label image is converted to a 32-bit float, which is displayed as raw values with a colour LUT. Unfortunately, float loses integer precision for values higher than $2^{23}$, which dramatically reduces the possible label range to a maximum of 8 388 608. Eight million labels is still sufficient for most images up to around $8 \times 10^6 \times 10^4$ or $8 \times 10^{10}$ pixels, about

80 GB. Should users have a need to process larger images containing more particles than these limits, the float precision restriction could be worked around.

# 7. Conclusion

This report describes an optimized implementation of sequential region labelling that fully exploits computational hardware resources and achieves theoretically optimum linear ($O$(n)) or slightly better than linear scaling, providing connected components labelling of multi-GB images in a time scale of seconds to minutes instead of hours to days. It is provided as a working, easy-to-use package in BoneJ for ImageJ, as source code, and has an API for other developers to use.

Data accessibility. BoneJ source code and changes: https://github.com/bonej-org/BoneJ2. BoneJ release on Zenodo [31] doi:10.5281/zenodo.3726422. Test images and performance testing results doi:10.6084/m9.figshare.11860542. Test scripts doi:10.6084/m9.figshare.11860536.

Competing interests. M.D. was a member of the Editorial Board of Royal Society Open Science at the time of submission and was not involved in the assessment of this submission.

Funding. BoneJ2 infrastructure development was supported by a Wellcome Trust Biomedical Resource and Technology Development Grant (108442/Z/15/Z).

Acknowledgements. Thank you to Robert Haase (MPI-CBG) for discussions and encouragement and to Alessandro Felder (UCL) and Richard Domander (RVC) for help with BoneJ2 engineering. Steve Chai and Meiji Ma (ThermoFisher) advised on the use of Avizo. I would be pleased to credit the (so far) anonymous originator of the *s10up* image that was used for testing this algorithm.

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
