## [Peer Review File · Royal Society Open Science]

Review History

RSOS-200506.R0 (Original submission)

Review form: Reviewer 1

Is the manuscript scientifically sound in its present form?

Yes

Are the interpretations and conclusions justified by the results?

Yes

Is the language acceptable?

Yes

Do you have any ethical concerns with this paper?

No

Have you any concerns about statistical analyses in this paper?

No

Recommendation?

Accept as is

Comments to the Author(s)

This paper develops a multithreaded implementation of two-pass labelling. The overall paper is well-organized. It can be accepted.

Review form: Reviewer 2**Is the manuscript scientifically sound in its present form?**

Yes

Are the interpretations and conclusions justified by the results?

Yes

Is the language acceptable?

Yes

Do you have any ethical concerns with this paper?

No

Have you any concerns about statistical analyses in this paper?

Yes

Recommendation?

Reject

Comments to the Author(s)

My review is in 2 parts: architecture point of view and algorithmic point of view

architecture

the implementation is very inefficient because of Java choice (instead of C or C++)

But the main problem is the scalability: the code has a constant weak scalability even when the number of thread is more than then number of core.

Because of the Hyper Threading technology within Intel Processor, the code can scale when the number of threads is the double od the number of cores, but not after.

That focus on the fact that java code is very inefficient

For 2-MB data, the speedup is correct, up to 4 thread and up to 16 threads for 1-GB data and 40 threads for 6.5 GB-data

If we compute the number of CPU cycle per pixel: that is the execution time multiplied by the CPU frequency and divided by the size of data, we approximatively get the same resultats whatever the data size is: with 1 thread $cpp = 240$ for 2 MB and with 40 threads, $cpp=51$. That represents a speedup of 2.9 and 4.75.

But the execution time are definitively slow: 240 cycles/point should be compared to 10 cycles/point (or less) for the best mono-threaded algorithms in C (see YACCLAB library)

algorithm

The bibliography is about experiments in various fields of research, there is neither recent citation about Connected Component Labeling nor comparisons with existing implementations (YACCLAB or benchmark in papers)

Review form: Reviewer 3

Is the manuscript scientifically sound in its present form?

Yes

Are the interpretations and conclusions justified by the results?

Yes

Is the language acceptable?

Yes

Do you have any ethical concerns with this paper?

No

Have you any concerns about statistical analyses in this paper?

No

Recommendation?

Accept with minor revision (please list in comments)

Comments to the Author(s)

This is an interesting and important contribution of connected component analysis. It is of relevance in many fields, e.g. in bioscience of bone. The paper is well written and the methods presented are already released as part of the BoneJ package of ImageJ.

The methods are compare with Avizo, which is a suitable reference. It would be worth to contrast the method to other proposed solutions, e.g. Mader, K. S.; Schneider, P.; Müller, R.; Stampanoni, M., A quantitative framework for the 3D characterization of the osteocyte lacunar system. Bone 2013, 57 (or at least cite this paper that in part builds on connected component analysis of the characterization of osteocyte lacunae in bone), especially in terms of what is calculated on the connected components after they have been found.

Further, I wonder how the method compares to other implementations, e.g. <https://github.com/seung-lab/connected-components-3d?>

In line 269 (page 9 of 11): shouldn't "1024 GB: 37 s)," be "1024 MB: 37 s);"?

Decision letter (RSOS-200506.R0)

Dear Dr Doube

The Editors assigned to your paper RSOS-200506 "Multithreaded two-pass connected components labelling and particle analysis in ImageJ" have made a decision based on their reading of the paper and any comments received from reviewers.

Firstly, our sincere apologies for the delays incurred during peer-review; owing to the COVID-19 crisis, several Editors were unavailable, and it was unusually difficult to secure referees.

Regrettably, in view of the reports received, the manuscript has been rejected in its current form. However, a new manuscript may be submitted which takes into consideration these comments.

We invite you to respond to the comments supplied below and prepare a resubmission of your manuscript. Below the referees' and Editors' comments (where applicable) we provide additional requirements. We provide guidance below to help you prepare your revision.

Please note that resubmitting your manuscript does not guarantee eventual acceptance, and we do not generally allow multiple rounds of revision and resubmission, so we urge you to make every effort to fully address all of the comments at this stage. If deemed necessary by the Editors, your manuscript will be sent back to one or more of the original reviewers for assessment. If the original reviewers are not available, we may invite new reviewers.

Please resubmit your revised manuscript and required files (see below) no later than 21-Mar-2021. Note: the ScholarOne system will 'lock' if resubmission is attempted on or after this deadline. If you do not think you will be able to meet this deadline, please contact the editorial office immediately.

Please note article processing charges apply to papers accepted for publication in Royal Society Open Science (<https://royalsocietypublishing.org/rsos/charges>). Charges will also apply to papers transferred to the journal from other Royal Society Publishing journals, as well as papers submitted as part of our collaboration with the Royal Society of Chemistry (<https://royalsocietypublishing.org/rsos/chemistry>). Fee waivers are available but must be requested when you submit your manuscript (<https://royalsocietypublishing.org/rsos/waivers>).

Thank you for submitting your manuscript to Royal Society Open Science and we look forward to receiving your resubmission. If you have any questions at all, please do not hesitate to get in touch.

on behalf of the Associate Editor and Professor Marta Kwiatkowska (Subject Editor)
openscience@royalsociety.org

Reviewer comments to Author:
Reviewer: 1
Comments to the Author(s)

This paper develops a multithreaded implementation of two-pass labelling. The overall paper is well-organized. It can be accepted.

Reviewer: 2
Comments to the Author(s)

My review is in 2 parts: architecture point of view and algorithmic point of view

architecture

the implementation is very inefficient because of Java choice (instead of C or C++)

But the main problem is the scalability: the code has a constant weak scalability even when the number of thread is more than then number of core.

Because of the Hyper Threading technology within Intel Processor, the code can scale when the number of threads is the double od the number of cores, but not after.

That focus on the fact that java code is very inefficient

For 2-MB data, the speedup is correct, up to 4 thread and up to 16 threads for 1-GB data and 40 threads for 6.5 GB-data

If we compute the number of CPU cycle per pixel: that is the execution time multiplied by the CPU frequency and divided by the size of data, we approximatively get the same resultats whatever the data size is: with 1 thread cpp = 240 for 2 MB and with 40 threads, cpp=51. That represents a speedup of 2.9 and 4.75.

But the execution time are definitively slow: 240 cycles/point should be compared to 10 cycles/point (or less) for the best mono-threaded algorithms in C (see YACCLAB library)

algorithm

The bibliography is about experiments in various fields of research, there is neither recent citation about Connected Component Labeling nor comparisons with existing implementations (YACCLAB or benchmark in papers)

Reviewer: 3

Comments to the Author(s)

This is an interesting and important contribution of connected component analysis. It is of relevance in many fields, e.g. in bioscience of bone. The paper is well written and the methods presented are already released as part of the BoneJ package of ImageJ.

The methods are compare with Avizo, which is a suitable reference. It would be worth to contrast the method to other proposed solutions, e.g. Mader, K. S.; Schneider, P.; Müller, R.; Stampanoni, M., A quantitative framework for the 3D characterization of the osteocyte lacunar system. Bone 2013, 57 (or at least cite this paper that in part builds on connected component analysis of the characterization of osteocyte lacunae in bone), especially in terms of what is calculated on the connected components after they have been found.

Further, I wonder how the method compares to other implementations, e.g. <https://github.com/seung-lab/connected-components-3d?>

In line 269 (page 9 of 11): shouldn't "1024 GB: 37 s)," be "1024 MB: 37 s),"?"

===PREPARING YOUR MANUSCRIPT===

===PREPARING YOUR REVISION IN SCHOLARONE===

<https://royalsociety.org/journals/authors/author-guidelines/#supplementary-material> to include a suitable title and informative caption. An example of appropriate titling and captioning may be found at https://figshare.com/articles/Table_S2_from_Is_there_a_trade-off_between_peak_performance_and_performance_breadth_across_temperatures_for_aerobic_sc_ope_in_teleost_fishes_/3843624.

Author's Response to Decision Letter for (RSOS-200506.R0)

See Appendix A.

RSOS-201784.R0

Review form: Reviewer 1

Is the manuscript scientifically sound in its present form?

Yes

Are the interpretations and conclusions justified by the results?

Yes

Is the language acceptable?

Yes

Do you have any ethical concerns with this paper?

No

Have you any concerns about statistical analyses in this paper?

No

Recommendation?

Accept as is

Comments to the Author(s)

I have no more comments. Thank you.

Review form: Reviewer 2

Is the manuscript scientifically sound in its present form?

Yes

Are the interpretations and conclusions justified by the results?

Yes

Is the language acceptable?

Yes

Do you have any ethical concerns with this paper?

No

Have you any concerns about statistical analyses in this paper?

No

Recommendation?

Accept as is

Comments to the Author(s)

This is a great piece of engineering work (for a Java code).

It is still outperformed by State of the Art algorithm

The only reason to accept this paper is that the java code is open (and increase ImageJ) and so cheaper than a commercial software.

Decision letter (RSOS-201784.R0)

Dear Dr Doube,

I am pleased to inform you that your manuscript entitled "Multithreaded two-pass connected components labelling and particle analysis in ImageJ" is now accepted for publication in Royal Society Open Science.

You can expect to receive a proof of your article in the near future. Please contact the editorial office (openscience@royalsociety.org) and production office (openscience_proofs@royalsociety.org) to let us know if you are likely to be away from e-mail contact -- if you are going to be away, please nominate a co-author (if available) to manage the proofing process, and ensure they are copied into your email to the journal.

Royal Society Open Science operates under a continuous publication model. Your article will be published as soon as it is ready for publication, and this will be the final version of the paper. As such, it can be cited immediately by other researchers. As the issue version of your paper will be the only version to be published I would advise you to check your proofs thoroughly as changes cannot be made once the paper is published.

on behalf of Professor Bart de Moor (Associate Editor) and Marta Kwiatkowska (Subject Editor)
openscience@royalsociety.org

Reviewer comments to Author:
Reviewer: 1

Comments to the Author(s)
I have no more comments. Thank you.

Reviewer: 2

Comments to the Author(s)
This is a great piece of engineering work (for a Java code).
It is still outperformed by State of the Art algorithm
The only reason to accept this paper is that the java code is open (and increase ImageJ) and so cheaper than a commercial software.

Appendix A

Dear Professor Kwiatkowska,

Thank you for taking the time to consider this manuscript, which describes an implementation of multithreaded two-pass connected components labelling in Java for ImageJ that far exceeds the performance of a popular commercial implementation in C++ ('Labeling' in Avizo). Its main value is in its convenience, which is achieved not only through very good real-world performance that scales well on very large images (as $O(n)$), but also through ease of installation and operation, so the total time cost to users is minimised. Please see below my response to the reviews, and the modified text. Note that while the first submission was in review I made some changes that increased speed by 10-25%, so all the performance measurements have been repeated.

Best regards,

Dr Michael Doube
Hong Kong, October 2020

Reviewer 1:

This paper develops a multithreaded implementation of two-pass labelling. The overall paper is well-organized. It can be accepted.

Thank you for your positive comment.

Reviewer: 2

My review is in 2 parts: architecture point of view and algorithmic point of view

architecture

the implementation is very inefficient because of Java choice (instead of C or C++)

Java having poor performance is a commonly-held misconception. Java's array access approaches or beats that of C and C++, especially for sequential reads, and its command execution can match or exceed C and C++. It was the case about 25 years ago that C and C++ greatly outperformed Java. Improvements in the JRE and the just-in-time compiler producing processor-specific optimised code mean that since the early 2000s Java is similar to and in some cases faster than C and C++. See e.g. <http://scribblethink.org/Computer/javaCbenchmark.html>

Where Java can be slow is in creation and handling of some Objects, and this is to be avoided especially in inner loops (although C++ can suffer from similar limitations). As acknowledged in the Discussion (L290), creation and population of HashSets is the most expensive part of this implementation, and reducing it is likely to be the most rewarding in terms of further optimisation. While this manuscript was in its first round of review a c. 25% speed improvement was made by replacing Java's `HashSet<Integer>` with Eclipse Collections' primitive `IntHashSet`. **All the performance data are updated with results from the new code**, and a new figure (Fig 4) shows how the multithreaded first and second passes scale at $n < 1$, and the single-threaded collision resolution at $n = 1.03$. The advantage of using Collections in Java is that the collision recording array size does not have to be guessed and reserved ahead of time, nor suffers from buffer overflows as C++ arrays do.

Benchmarking this code against a commercial multithreaded implementation in C++ (Avizo 2020.2) shows that this Java implementation is about 15× faster. A main component of real-world efficiency is how long it takes the user to get results from data, which may include raising a purchase order (for Avizo, this is substantial), then downloading, installing and operating the software. Time spent opening images and manipulating menus and other GUI elements, or even understanding how to install and run software, is non-trivial. Implementation in Java means that it has to be written only

once (saving programmer time) and can run wherever there is a JRE, without the need for maintaining several native versions. Most normal users will not complete difficult setup and running steps that might include learning Git and/or bash to clone the code, compiling with gcc or running a Python script. This implementation, as part of BoneJ, is very easy to install and operate with the express purpose of saving users time (it can also be operated via scripts, for users who are so inclined).

I added text explaining these points to lines:

L275 Completion times were approximately 15× less than Avizo's multithreaded Labeling module, (1 GB: 9.5 – 11s vs. 165 s; 6.5 GB: 65 - 72s vs. 1220 s).

L310 Alternative implementations exist in programming languages that can be more efficient than Java at accessing large amounts of array data, such as C and C++ [23,26], or that store particle label collisions in primitive arrays that can be faster to create and access than the Java (Eclipse) Collections used here. This Java implementation far exceeds the performance of the commercial C++ implementation included in Avizo's Labeling module, with further optimisations still possible such as a kernel decision tree (after [23,26]). A major goal of this project was to produce an implementation that is trivial to install and run for users of the popular ImageJ platform [15]. The Fiji distribution makes it convenient for users to install and operate the plugin, and ImageJ is written in Java, hence the design decision to implement this algorithm in Java.

But the main problem is the scalability: the code has a constant weak scalability even when the number of thread is more than then number of core.

As explained in the text, the algorithm scales very well, at or better than the theoretical optimum of $O(n)$, (i.e. time to complete is linearly proportional to image size in pixels, for any given number of threads). Figure 4 shows that the multithreaded first and second passes scale better than linearly, while the collision resolution step (which runs in only a single thread due to its design) is slightly more at $n = 1.03$. It hits the *memory wall* at about 10-16 threads (on a 20-core, 40-thread dual CPU machine), which relates to hardware limitation and not to poor design of the algorithm itself. Providing more memory parallelism should increase the throughput of the implementation.

Because of the Hyper Threading technology within Intel Processor, the code can scale when the number of threads is the double od the number of cores, but not after.

Agreed, and this is precisely what was tested: up to 40 Java threads were created to run on 40 CPU threads on a 20-core machine (each core has 2 threads on the test machine, as described in materials and methods L220). Here, the performance scaling stops increasing while Java threads is fewer than CPU cores (and less than half the available CPU threads); this is presumably due to the memory wall, as described in the Discussion, L300. In other words, memory bandwidth is saturated before CPU cycles are fully used, which is not surprising because very little calculation is performed for each pixel read.

That focus on the fact that java code is very inefficient

As explained above, it is not, at least not necessarily.

For 2-MB data, the speedup is correct, up to 4 thread and up to 16 threads for 1-GB data and 40 threads for 6.5 GB-data

I am not sure what point the reviewer is trying to make here.

If we compute the number of CPU cycle per pixel: that is the execution time multiplied by the CPU frequency and divided by the size of data, we approximatively get the same resultats whatever the data size is: with 1 thread $c_{pp} = 240$ for 2 MB and with 40 threads, $c_{pp}=51$. That represents a speedup of 2.9 and 4.75.

An alternative approach to this problem is expressing it as pixels per second, which relates inversely to cpp. Ideally, pixels per second should be constant as image size increases, and increase linearly with the number of threads. What actually happens is that pixels per second relates unpredictably to data size, with the fastest being the 128 MB image which has 512-pixel sides, the slowest being the 2 MB image and the others in-between, which I suspect is to do with the way contiguous chunks of memory are passed from RAM to the CPU cache or between CPU cache levels. The ideal thread scaling does not occur due to memory bandwidth limiting data throughput to CPU. See the `performance_test_table.ods` in the online data for the full breakdown.

But the execution time are definitively slow: 240 cycles/point should be compared to 10 cycles/point (or less) for the best mono-threaded algorithms in C (see YACCLAB library).

Thank you for the pointer to YACCLAB. In a single thread, the 128 MB test image is processed at a rate of 24 M pixels per second in a single thread of a 2.2 GHz processor, leading to 92 cpp. As noted, the speedup with increasing threads is not linear due to memory bound effects. Maximum pixel throughput is 140 Mpixels per second (on 40 threads), having a similar completion time to a 15 cpp single-threaded implementation. For the user, that is nearly as good as it gets (and there is still room for improvement).

algorithm

The bibliography is about experiments in various fields of research, there is neither recent citation about Connected Component Labeling nor comparisons with existing implementations (YACCLAB or benchmark in papers)

Thank you for pointing to these, they are very helpful and some more recent literature has been cited. The implementation is now benchmarked against a commercial implementation included in Avizo, a commonly used commercial software for 3D volumetric analysis (L272).

Reviewer: 3

Comments to the Author(s)

This is an interesting and important contribution of connected component analysis. It is of relevance in many fields, e.g. in bioscience of bone. The paper is well written and the methods presented are already released as part of the BoneJ package of ImageJ.

Thank you for your supportive comments.

The methods are compare with Avizo, which is a suitable reference. It would be worth to contrast the method to other proposed solutions, e.g. Mader, K. S.; Schneider, P.; Müller, R.; Stampanoni, M., A quantitative framework for the 3D characterization of the osteocyte lacunar system. Bone 2013, 57 (or at least cite this paper that in part builds on connected component analysis of the characterization of osteocyte lacunae in bone), especially in terms of what is calculated on the connected components after they have been found.

Thanks for the reminder about that paper. Once the labels are found, any number of specialised measurements can be made on them, such as those mentioned above about the osteocyte lacunocanalicular system. I added a paragraph explaining this at L320:

The redesign of the code to be more modular addresses concerns expressed in previous work, such as Mader et al. (2013) who complained that the prior implementation in BoneJ “*lacks the flexibility to perform a number of different analyses, necessary for large-scale studies and detailed analysis of distribution, alignment, and other osteocyte lacunar measures*” [28]. Once the label image is created, any range of analyses could be performed on the labels, by adding analytic methods to BoneJ’s new `ParticleAnalysis` class, by obtaining the label image programatically by calling the

`ConnectedComponents.run()` method from client code, or by running custom analysis code such as a Python script or ImageJ plugin on the label image that is displayed in the ImageJ GUI. The label image could also be saved and opened in other software.

Further, I wonder how the method compares to other implementations, e.g.
<https://github.com/seung-lab/connected-components-3d?>

Thank you for the link. The new implementation shares a few features with the code linked above, which also gave some ideas for future improvements (the neighbourhood decision tree in particular), and some handy recent references that I have cited e.g. Wu et al. I installed the software to see how it performs, but failed to see how to make it operate on my images due to a lack of documentation for a user new to Python as I am, and gave up.

In line 269 (page 9 of 11): shouldn't "1024 GB: 37 s)," be "1024 MB: 37 s),"?
Yes, thank you, corrected.